

# A spatially variant high-order variational model for Rician noise removal

Tran Dang Khoa Phan

Faculty of Electronics and Telecommunication Engineering, University of Science and Technology - The University of Danang, Danang, Vietnam

## ABSTRACT

Rician noise removal is an important problem in magnetic resonance (MR) imaging. Among the existing approaches, the variational method is an essential mathematical technique for Rician noise reduction. The previous variational methods mainly employ the total variation (TV) regularizer, which is a first-order term. Although the TV regularizer is able to remove noise while preserving object edges, it suffers the staircase effect. Besides, the adaptability has received little research attention. To this end, we propose a spatially variant high-order variational model (SVHOVM) for Rician noise reduction. We introduce a spatially variant TV regularizer, which can adjust the smoothing strength for each pixel depending on its characteristics. Furthermore, SVHOVM utilizes the bounded Hessian (BH) regularizer to diminish the staircase effect generated by the TV term. We develop a split Bregman algorithm to solve the proposed minimization problem. Extensive experiments are performed to demonstrate the superiority of SVHOVM over some existing variational models for Rician noise removal.

# INTRODUCTION

Magnetic resonance (MR) images have been widely used in medical imaging. Due to the thermal noise caused by patients during the scan process (*Nowak, 1999*; *Aja-Fernández & Vegas-Sánchez-Ferrero, 2016*), the MR images are inevitably degraded. The noises in the MR images have negative impacts on various tasks of medical image processing and analysis, such as classification, segmentation, visualization (*Aja-Fernández & Vegas-Sánchez-Ferrero, 2016*). Hence, noise removal is the fundamental task for processing MR images.

It was shown that the noises in the MR images can be modeled by the Rician distribution (*Henkelman, 1985*; *Aja-Fernández & Vegas-Sánchez-Ferrero, 2016*). The task of Rician noise removal refers to estimating the clean MR image from a noisy one. Since the Rician noise is signal-dependent, it is a great challenge to denoise the clean MR image. Next, the previous works that address the problem of Rician noise removal are reviewed. First, several denoising methods based on statistics are presented. In (*Henkelman, 1985*; *McGibney & Smith, 1993*; *Bernstein, Thomasson & Perman, 1989*), the first and second moments of the Rician distribution were employed to estimate the clean MR images. Using the local sample statistics, *Aja-Fernández & Vegas-Sánchez-Ferrero (2016)* derived a closed-form

Corresponding author
Tran Dang Khoa Phan,
ptdkhoa@dut.udn.vn

solution of the linear minimum mean square error (LMMSE) estimator for the Rician distribution. Many variants of the non-local means (NLM) algorithm have been developed for Rician noise reduction to enhance the signal-to-noise ratio and the computational efficiency (*Manjón et al., 2008*; *Baselice et al., 2019*; *Granata, Amato & Alfano, 2019*; *Phan, 2018*; *Chen et al., 2020*; *Sharma & Chaurasia, 2021*; *Zhang et al., 2021*). In recent years, learning-based methods have been applied to Rician noise removal. In *You et al. (2019)* proposed a deep convolutional neural network (CNN) for non-blind and blind Rician denoising. In *Manjón & Coupe (2018)* studied a two-stage approach which combines the overcomplete patch-based CNN and the NLM filter to robustly reduce noises in MR images. *Xie et al. (2020)* presented a denoising network based on CNN with dilated convolutions and residual blocks.

Along with the above mentioned approaches, the variational method is a crucial mathematical technique for Rician noise removal. A variational model usually has two terms: the data fidelity and the regularizer. The first term measures the fidelity to the noisy image and the latter term poses constraints for the solution. One of the widely used regularizer is the total variation (TV), which was proposed by *Rudin, Osher & Fatemi (1992)* for Gaussian denoising. The TV regularizer is able to reduce noise while maintaining object edges. Based on the framework of maximum a posterior (MAP) estimates, *Getreuer, Tong & Vese (2011)* proposed a TV-based variational model with the data fidelity term derived from the Rician probability distribution. This MAP model, however, is non-convex, and thus its solution depends on the initialization and numerical methods. To address this drawback, *Getreuer, Tong & Vese (2011)* approximated the MAP model by a convexified one. The author investigated the $\ell^2$ and Sobolev $H^1$ gradient descents for the MAP model and the split Bregman for the convexified model. Considering the statistical property of the Rician noise, *Chen & Zeng (2015)* added a quadratic term into the non-convex MAP model to obtain a strictly convex model. In *Yuan (2018)* introduced a convex gradient data fidelity term into the MAP model. Besides, the noise level is iteratively estimated to improve the denoised results. Unlike the MAP-based approach, *Liu, Chang & Duan (2022)* proposed a non-linear model which consists of quadratic terms, a constraint on the field of spheres, and a TV regularizer. Other variants of variational models for Rician denoising can be found in *Liu et al. (2014)*, *Chen et al. (2018)*, *Lu et al. (2019)*, *Pankaj, Govind & Narayanankutty (2021)* and *Phan (2022)*.

In this article, variational models are mainly investigated. The above overview shows that most of the existing variational methods focus on the fidelity term. Meanwhile, the regularization term attracts less attention. The TV regularizer is still widely used for Rician noise removal. Although the TV term is capable of removing noise and preserving edges, it produces the staircase effect, that is, the restored image appears jagged. Besides, previous works were less concerned with the adaptability to the characteristics of pixels. Namely, the regularization strength of the TV term is the same for all pixels. In this article, a spatially variant high-order variational model (SVHOVM) for Rician noise removal is presented. The author introduces a spatially variant TV (SVTV) regularizer which can control its smoothing strength depending on whether pixels are in flat regions or at object edges. Besides, the proposed model applies the bounded Hessian (BH) regularizer to reduce the

staircase effect generated by TV. An efficient split Bregman algorithm is developed to solve the proposed model. The proposed model is evaluated on a large dataset in comparison with several existing variational models for Rician noise removal. Experimental results show the superiority of SVHOVM in Rician denoising.

The main contributions include the following:

- A novel variational model for Rician noise removal is proposed. Particularly, the common TV regularizer is modified such that it becomes spatially variant according to the characteristics of pixels. The BH regularizer, which is a high-order term, is utilized to enhance the denoising results.
- An efficient split Bregman algorithm is developed to solve the proposed problem.
- Extensive experiments are conducted to discuss the effects of the parameters of SVHOVM and to evaluate its performance. Experimental results show that the proposed model outperforms some existing variational models for Rician noise reduction.

The rest of the article is organized as follows. In Section 'Preliminary and related works', some preliminaries and a brief overview of related works are presented. The proposed model is described in Section 'Proposed model'. In Section 'Numerical implementation', a split Bregman algorithm for solving the proposed problem is presented. Section 'Experimental results' discusses experimental results.

## PRELIMINARY AND RELATED WORKS

### Preliminary

The MR imaging systems use quadrature detectors to produce two- or three-dimensional complex data. The raw MR data is always perturbed by Gaussian noise. The complex representation of the raw MR data is given by

$$\mathcal{F} = \mathcal{F}_R + i\mathcal{F}_I = u + \eta_1 + i\eta_2, \tag{1}$$

where $\mathcal{F}_R$ and $\mathcal{F}_I$ are the real and imaginary parts of the raw MR data $\mathcal{F}$; $u \in \mathbb{R}^{p \times q}$ is the true amplitude of the noise-free image; $\eta_1$ and $\eta_2 \in \mathbb{R}^{p \times q}$ are Gaussian noise with zero mean and standard deviation $\sigma$.

For clinical analysis, the magnitude MR images are often used. Mathematically, the magnitude MR image is computed by

$$f = \sqrt{(u + \eta_1)^2 + \eta_2^2}. \tag{2}$$

Since the magnitude MR images are obtained by the non-linear transformation, the distribution of the overall noises for the magnitude MR image is no longer Gaussian. It was shown in *Henkelman (1985)*, *Aja-Fernández, Alberola-López & Westin (2008)* that the noises in the magnitude MR images have the Rician distribution which is given by

$$\mathbb{P}(f|u) = \frac{f}{\sigma^2} \exp(-\frac{u^2 + f^2}{2\sigma^2}) I_0(\frac{uf}{\sigma^2}), \tag{3}$$

where $I_0$ is the modified Bessel function of the first kind with order zero (*Bowman, 2012*). The form of the modified Bessel function of the first kind with real order $v$ are given by

$$I_v(z) = (\frac{1}{2}z)^v \sum_{k=0}^{\infty} \frac{(\frac{1}{4}z^2)^k}{k!\Gamma(v+k+1)}, \tag{4}$$

with $\Gamma(n) = (n-1)!$ is the gamma function; $v \in \mathbb{R}$.

## Related works

The goal of Rician noise removal is to estimate the noise-free image $u$ from the noisy magnitude MR image $f$. Most of variational models utilize the MAP approach to estimate $u$ by maximizing a posterior given $f$, that is $\tilde{u} = \arg\max_u \mathbb{P}(u|f)$. In *Getreuer, Tong & Vese (2011)*, they applied the Bayes's rule to derive the MAP model as

$$\arg\min_u \left\{ \frac{1}{2\sigma^2} \int_\Omega u^2 dx - \int_\Omega \log I_0(\frac{uf}{\sigma^2}) dx + \alpha \int_\Omega |\nabla u| dx \right\}, \tag{5}$$

where the first two terms form the data fidelity, which is derived from (3) using the MAP framework; the last term is the TV of $u$; $\alpha$ is a non-negative regularization parameter; $\nabla$ is the gradient operator; $\Omega$ is the image domain.

Since the MAP model is non-convex, *Getreuer, Tong & Vese (2011)* approximated its data fidelity term by a convex function as follows

$$G_\sigma(u) = \begin{cases} H_\sigma(u), & \text{if } u \geq c\sigma, \\ H_\sigma(c\sigma) + H'_\sigma(c\sigma)(u-c\sigma), & \text{if } u \leq c\sigma, \end{cases} \tag{6}$$

where

$$H_\sigma(u) = \frac{u^2}{2\sigma^2} - \log I_0(\frac{uf}{\sigma^2}), \tag{7}$$

$$H'_\sigma(u) = \frac{u}{\sigma^2} - \frac{f}{\sigma^2} A(\frac{uf}{\sigma^2}), \tag{8}$$

with $A(\cdot)$ is the cubic rational polynomial approximation of $I_1(\cdot)/I_0(\cdot)$ with $I_1(\cdot)$ being the modified Bessel function of the first kind with first order; $c = 0.8426$.

By exploring the statistical property of the Rician distribution, *Chen & Zeng (2015)* added the quadratic term into the MAP model to obtain a strictly convex model as

$$\arg\min_u \left\{ E(u) = \frac{1}{2\sigma^2} \int_\Omega u^2 dx - \int_\Omega \log I_0(\frac{uf}{\sigma^2}) dx + \frac{1}{\sigma} \int_\Omega (\sqrt{u} - \sqrt{f})^2 dx + \alpha \int_\Omega |\nabla u| dx \right\}. \tag{9}$$

Following the similar idea, *Yuan (2018)* added a convex gradient data fidelity into the MAP model as

$$\arg\min_u \left\{ \frac{1}{2\sigma^2} \int_\Omega u^2 dx - \int_\Omega \log I_0(\frac{uf}{\sigma^2}) dx + \frac{1}{\sigma} \int_\Omega (\nabla u - \nabla f)^2 dx + \alpha \int_\Omega |\nabla u| dx \right\}. \tag{10}$$

For brevity, the following notations are used: GTV for the convexified model derived by *Getreuer, Tong & Vese (2011)* (Eqs. (5)–(8)); CZ for the model of *Chen & Zeng (2015)* (Eq.(9)); *Yuan (2018)* for the model proposed by Yuan (Eq. (10)).

## PROPOSED MODEL

As described in 'Introduction', existing variational methods for Rician noise removal mainly utilize TV. Besides, the adaptability received less research attention. To this end, the author proposes a spatially variant high-order variational model (SVHOVM) for Rician denoising. The spatially variant TV (SVTV) and the bounded Hessian (BH) regularizers are introduced by minimizing the following minimization functional

$$\arg\min_u \left\{ \frac{1}{2\sigma^2}\|u\|_2^2 - \left\langle \log I_0(\frac{uf}{\sigma^2}), 1 \right\rangle + \frac{1}{\sigma}\|\sqrt{u} - \sqrt{f}\|_2^2 + \|\alpha(f)\nabla u\|_1 + \beta\|\nabla^2 u\|_1 \right\}, \quad (11)$$

where $<\cdot,\cdot>$ denotes the Euclidean inner product; $\|\cdot\|_1$ and $\|\cdot\|_2$ stand for the $\ell^1$- and $\ell^2$-norms, respectively; $\|\alpha(f)\nabla u\|_1$ is the SVTV regularizer with $\alpha(\cdot)$ being the weighting function; $\|\nabla^2 u\|_1$ is the BH regularizer where $\nabla^2$ denotes the Hessian operators; $\beta$ are non-negative regularization parameters.

The BH regularizer is exploited to remove the side effect produced by the TV term. In *Papafitsoros & Schönlieb (2014)* showed that the BH regularizer is able to remedy the staircase effect and to preserve structural details. The SVTV term is the common TV regularizer weighted by the function $\alpha(\cdot)$ for each pixel. The weighting function is defined as

$$\alpha(f) = \frac{\alpha_0}{\sqrt{1 + (\frac{|\nabla G_\omega * f|}{\kappa})^2}}, \quad (12)$$

where $\alpha_0$ is a non-negative parameter; $G_\omega$ stands for the Gaussian filter with zero mean and standard deviation $\omega$; $\kappa$ denotes a contrast parameter; "$*$" represents the convolution operator.

The weighting function can adaptively manipulate the smoothing strength of the TV regularizer. Its values vary depending on the image gradients of pixels. Namely, for a fixed $\kappa$, in flat regions where $|\nabla G_\omega * f| < \kappa$, the weighting function is large, which means a strong noise reduction. In contrast, at object edges where $|\nabla G_\omega * f| > \kappa$, the weighting function is small, which indicates the edge preservation. Thus, the weighting function is effective in reducing noise while maintaining object edges.

## NUMERICAL IMPLEMENTATION

In this section, a split Bregman algorithm is developed to solve the proposed problem (11). Following (*Goldstein & Osher, 2009*), two auxiliary variables are introduced to obtain the following constrained problem:

$$\arg\min_{u,d,z} \left\{ \frac{1}{2\sigma^2}\|u\|_2^2 - \left\langle \log I_0(\frac{uf}{\sigma^2}), 1 \right\rangle + \frac{1}{\sigma}\|\sqrt{u} - \sqrt{f}\|_2^2 + \|\alpha(f)d\|_1 + \beta\|z\|_1 \right\} \quad (13)$$

such that $d = \nabla u, z = \nabla^2 u,$

where $d$ and $z$ are auxiliary variables.

Applying the Bregman iteration gives the following unconstrained problem:

$$\arg\min_{u,d,z}\left\{\frac{1}{2\sigma^2}\|u\|_2^2-\left\langle\log I_0(\frac{uf}{\sigma^2}),1\right\rangle+\frac{1}{\sigma}\|\sqrt{u}-\sqrt{f}\|_2^2+\|\alpha(f)d\|_1+\beta\|z\|_1+\right.$$
$$\left.\frac{\theta_1}{2}\|d-\nabla u-b_1\|_2^2+\frac{\theta_2}{2}\|z-\nabla^2 u-b_2\|_2^2\right\}, \tag{14}$$

where $b_1$ and $b_2$ are the Bregman iteration variables; $\theta_1$ and $\theta_2$ are the penalty parameters.

The problem Eq. (14) is solved by an alternating direction method (*Gabay & Mercier, 1976*; *Bertsekas, 2014*). In each step, either $u$, $d$ or $z$ is minimized while keeping other variables fixed. With $u$ and $z$ fixed, the $d$-subproblem is obtained as:

$$\arg\min_d\left\{\|\alpha(f)d\|_1+\frac{\theta_1}{2}\|d-\nabla u^k-b_1^k\|_2^2\right\}, \tag{15}$$

which has the following solution:

$$d^{k+1}=\frac{\nabla u^k+b_1^k}{|\nabla u^k+b_1^k|}\max(|\nabla u^k+b_1^k|-\frac{\alpha(f)}{\theta_1},0), \tag{16}$$

Similarly, the $z$-subproblem and its solution are given by

$$\arg\min_z\left\{\beta\|z\|_1+\frac{\theta_2}{2}\|z-\nabla^2 u^k-b_2^k\|_2^2\right\}, \tag{17}$$

$$z^{k+1}=\frac{\nabla^2 u^k+b_2^k}{|\nabla^2 u^k+b_2^k|}\max(|\nabla^2 u^k+b_2^k|-\frac{\beta}{\theta_2},0). \tag{18}$$

By fixing $d$ and $z$, the $u$-subproblem is obtained as

$$\arg\min_u\left\{\frac{1}{2\sigma^2}\|u\|_2^2-\left\langle\log I_0(\frac{uf}{\sigma^2}),1\right\rangle+\frac{1}{\sigma}\|\sqrt{u}-\sqrt{f}\|_2^2+\frac{\theta_1}{2}\|d^{k+1}-\nabla u-b_1^k\|_2^2+\right.$$
$$\left.\frac{\theta_2}{2}\|z^{k+1}-\nabla^2 u-b_2^k\|_2^2\right\}. \tag{19}$$

Let $E(u)$ denote the functional of (19). The Newton's method is applied to solve the $u$-subproblem (19) as follows:

$$u^{k+1}=u^k-\frac{E'(u^k)}{E''(u^k)}, \tag{20}$$

with

$$E'(u^k)=\frac{u^k}{\sigma^2}-\frac{f}{\sigma^2}\frac{I_1}{I_0}+\frac{1}{\sigma}(1-\sqrt{\frac{f}{u^k}})+\theta_1\mathrm{div}(d^{k+1}-\nabla u^k-b_1^k)-\theta_2\mathrm{div}^2(z^{k+1}-\nabla^2 u^k-b_2^k), \tag{21}$$

$$E''(u^k)=\frac{1}{\sigma^2}-\frac{f^2}{\sigma^4}[1-\frac{\sigma^2}{f}\frac{1}{u^k}\frac{I_1}{I_0}-(\frac{I_1}{I_0})^2]+\frac{\sqrt{f}}{2\sigma}\frac{1}{(u^k)^{3/2}}, \tag{22}$$

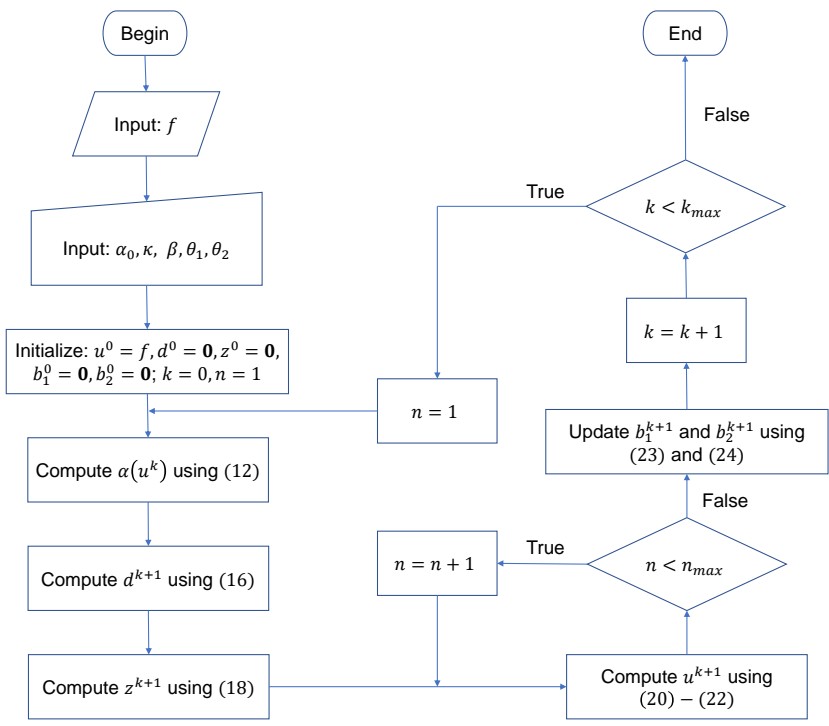

**Figure 1  The flowchart of the split Bregman algorithm for solving the proposed problem (11).** The parameters $k_{max}$ and $n_{max}$ denote the outer and inner iteration numbers, respectively.

where the variable $(uf/\sigma^2)$ of $I_0$ and $I_1$ is omitted for brevity.

Finally, the $b_1$ and $b_2$ variables are updated by:

$$b_1^{k+1} = b_1^k + \nabla u^{k+1} - d^{k+1}, \tag{23}$$

$$b_2^{k+1} = b_2^k + \nabla^2 u^{k+1} - z^{k+1}. \tag{24}$$

In summary, the denoised image is found by iteratively computing $u$, $d$ and $z$ via the sequence of Eqs. (16), (18), (20), (21), and (22). The number of iterations is used as the stopping criterion. It is worthy of note that the weighting function is iteratively refined by computing (12) using the restored image of the previous iteration. This refinement offers an enhanced weighting function, resulting in better denoised images. The overall split Bregman algorithm for solving the proposed problem (11) is summarized in Fig. 1.

## EXPERIMENTAL RESULTS

In this section, numerical experiments are conducted to discuss the affects of the parameters of SVHOVM and to evaluate the proposed model in comparison with existing variational methods for Rician noise removal. The experiments are performed on the IXI and SB datasets. The IXI dataset (Information eXtraction from Images: https://brain-development.org/ixi-dataset/.) contains real MR images. The SB dataset

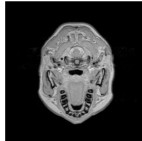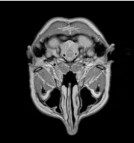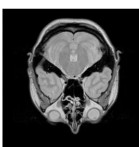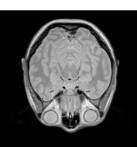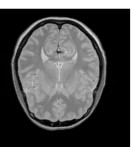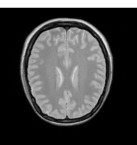

**Figure 2  Sample MR images.** Image source credit: IXI dataset, CC BY-SA 3.0 (https://brain-development.org/ixi-dataset/).

contains simulated MR images generated by BrainWeb (Simulated Brain Database: http://www.bic.mni.mcgill.ca/brainweb/.) (*Cocosco, 1997*; *Kwan, Evans & Pike, 1999*; *Kwan, Evans & Pike, 1996*; *Collins et al., 1998*). It consists of three-dimensional T1w, T2w, and PDw volumes of $181 \times 217 \times 181$ voxels with zero noise. The MR images are perturbed by Rician noise with three noise levels $\sigma = 5, 15$ and $25$. Sample MR images of the datasets are shown in Fig. 2.

The PSNR and SSIM (*Wang et al., 2004*) are used to measure the performance of models. Besides, visual quality is also employed for qualitative evaluation. Let $u$ and $\tilde{u}$ denote the noise-free and the denoised images. The PSNR and SSIM indices are defined as:

$$\text{PSNR} = 10\log_{10}\left(\frac{255^2}{\frac{1}{MN}\|\tilde{u} - u\|_2}\right), \tag{26}$$

$$\text{SSIM} = \frac{(2\mu_{\tilde{u}}\mu_u + c_1)(2\sigma_{\tilde{u},u} + c_2)}{(\mu_{\tilde{u}}^2 + \mu_u^2 + c_1)(\sigma_{\tilde{u}}^2 + \sigma_u^2 + c_2)}, \tag{27}$$

where $M$ and $N$ are the sizes of images; $\mu_{\tilde{u}}$, $\sigma_{\tilde{u}}$ and $\mu_u$, $\sigma_u$ are the means and standard deviations of $\tilde{u}$ and $u$, respectively; $c_1$ and $c_2$ are constants.

## Ablation study

In this section, numerical experiments are conducted to discuss the influence of various parameters on the proposed algorithm (Fig. 1). The regularization parameters $\alpha_0$ and $\beta$, the contrast parameter $\kappa$, and the inner iteration number $n_{max}$ are considered. Note that the inner iteration corresponds to the loop of the Newton's method (Eqs. 20–22).

### *Regularization parameters*

The effects of the parameters $\alpha_0$ and $\beta$ on the performance of SVHOVM are investigated. These parameters determine the weights of the SVTV and BH regularizers. One of these parameters is varied while keeping the other fixed.

The effects of $\alpha_0$ are shown in Figs. 3C–3E. The parameter $\beta$ is fixed to a low value (particularly, $\beta = 1$) in order to diminish the influence of this parameter. The parameter $\alpha_0$ is set to $1, 10$ and $20$ to show its effects. One can see that the parameter $\alpha_0$ controls the smoothness of denoising results. As $\alpha_0$ increases, more noise is reduced but the staircase effect becomes more obvious. Then, the influence of the parameter $\beta$ is examined. The parameter $\alpha_0$ is fixed by a large value (particularly, $\alpha_0 = 15$) in order to generate the staircase effect. The parameter $\beta$ is gradually increased to demonstrate its effects. Figures 3F–3H

**Peer**J Computer Science

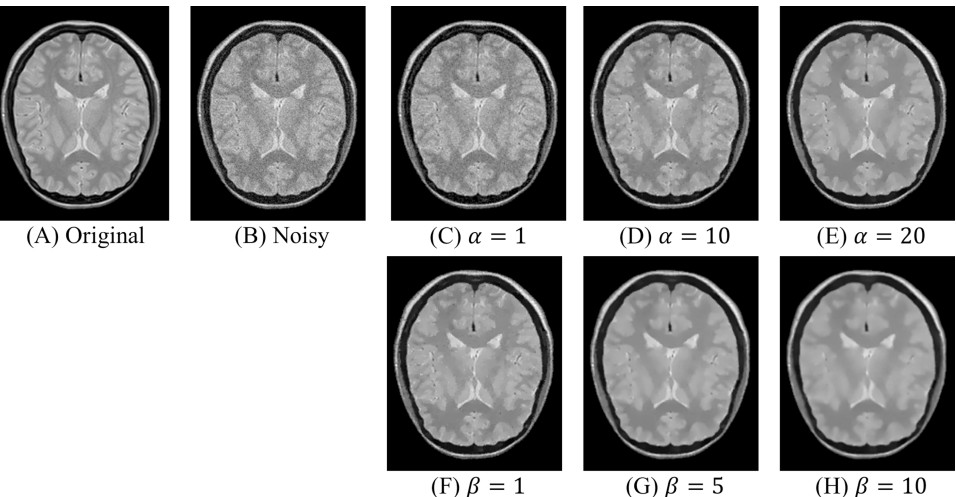

(A) Original     (B) Noisy     (C) $\alpha = 1$     (D) $\alpha = 10$     (E) $\alpha = 20$

(F) $\beta = 1$     (G) $\beta = 5$     (H) $\beta = 10$

**Figure 3** **The effects of the paramerters $\alpha_0$ and $\beta$ for an image of the IXI dataset at the noise level $\sigma = 15$: (C)–(E) for the fixed $\beta = 1$; (F)–(H) for the fixed $\alpha_0 = 15$.** Image source credit: IXI dataset, CC BY-SA 3.0 (https://brain-development.org/ixi-dataset/).

show that as $\beta$ gets larger, the BH regularizer diminishes the staircase effect more effectively, producing smooth transition between flat regions. However, the large values of $\beta$ result in blurred images. Thus, the regularization parameter $\beta$ should be not too large so that the artifacts generated by the TV term are diminished without generating any serious blur in the denoised images.

Next, the dependence of the PSNR measure on the parameters $\alpha_0$ and $\beta$ is considered. The parameter $\alpha_0$ is fixed at different values while $\beta$ is varied. Figure 4A shows that the PSNR values change in a continuous manner. For a fixed $\alpha_0$, the PSNR measures initially increase with the value of $\beta$, reaching the maximum value and then decreasing. When $\alpha_0$ increases, the PSNR measure rises to the global maximum, followed by a decrease. One can see from Fig. 4A that an optimal denoised result can be attained by alternatively adjusting the two regularization parameters $\alpha_0$ and $\beta$.

### Contrast parameter

Figure 4B shows the dependence of the weighting function $\alpha(\cdot)$ on the gradient magnitude for different values of the contrast parameter $\kappa$. As can be seen, the weighting function is monotonically decreasing with the increase of the gradient magnitude. As the gradient magnitude becomes larger, $\alpha(\cdot)$ goes to 0 and the strength of the TV term is down-weighted. Thus, the weighting function controls the regularization strength of the TV term. Figure 4B also demonstrates that the parameter $\kappa$ adjusts the range in which the weighting function receives low values. As $\kappa$ declines, this range is extended. It means that when $\kappa$ decreases, the more image details as well as noise are preserved. Therefore, an optimal value of $\kappa$ needs to be determined to achieve a balance between noise reduction and image detail preservation. The parameter $\kappa$ is set to 0.8 empirically.

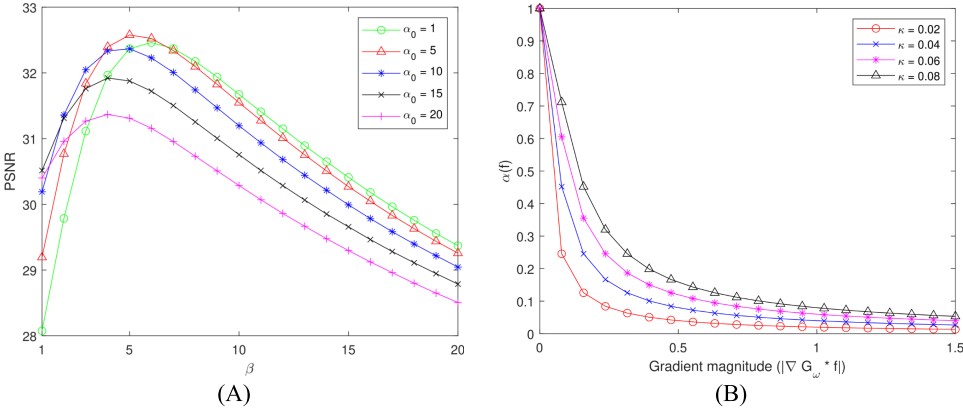

**Figure 4** The effects of the parameters $\alpha_0$, $\beta$, and $\kappa$. (A) The PSNR values under different $\alpha_0$ and $\beta$ settings at the noise level $\sigma = 15$; (B) The effect of the contrast parameter $\kappa$ on the weighting function $\alpha(\cdot)$.

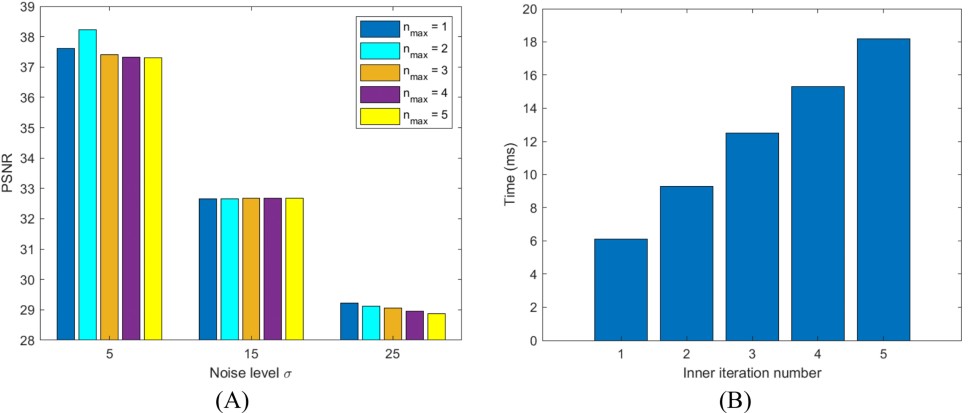

**Figure 5** The dependence of SVHOVM's performance on the inner iteration number for: (A) The PSNR values; (B) The average time per iteration. The outer iteration number $k_{max}$ is fixed by 500; the inner iteration number $n_{max}$ is set to $1, 2, .., 5$. The noise levels $\sigma = 5, 15$, and 25 are considered.

### Inner iteration number

The impacts of the inner iteration number $n_{max}$ on the performance of the proposed algorithm are examined. Figure 5A shows that the proposed algorithm is less sensitive to the number of inner iterations. Meanwhiles, the computational time per iteration of the proposed algorithm increases by about 25% on average when the number of inner loops is increased by one (Fig. 5B). Thus, one inner iteration is used in order to reduce the computational complexity of the proposed algorithm without affecting the quality of the denoised images.

### Comparative study

Next, SVHOVM is compared with some existing variational models for Rician noise removal. The models GTV, CZ, and Yuan are used as references. The regularization parameters of models are tuned using a simple alternating optimization method to achieve

   
**Table 1 Average PSNR and SSIM results of different methods on the IXI and SB datasets with different noise levels.** The values highlighted in bold represent the best results for each volume and noise level.

| Dataset | Method | $\sigma = 5$ | | $\sigma = 15$ | | $\sigma = 25$ | |
|---|---|---|---|---|---|---|---|
| | | PSNR | SSIM | PSNR | SSIM | PSNR | SSIM |
| IXI | GTV | 35.31 | 0.6458 | 29.13 | 0.5016 | 25.76 | 0.4179 |
| | CZ | 34.98 | 0.6312 | 28.74 | 0.4895 | 25.19 | 0.4003 |
| | Yuan | 35.34 | 0.6468 | 29.01 | 0.4927 | 25.45 | 0.41 |
| | SVHOVM | **36.23** | **0.6797** | **29.72** | **0.5066** | **26.09** | **0.5094** |
| SB | T1w | | | | | | |
| | GTV | 35.5 | 0.7459 | 29.54 | **0.65** | 25.98 | **0.5914** |
| | CZ | 35.37 | 0.734 | 28.96 | 0.5959 | 24.43 | 0.5186 |
| | Yuan | 35.48 | 0.7464 | 29.26 | 0.6356 | 25.48 | 0.5332 |
| | SVHOVM | **36.78** | **0.7758** | **30.4** | 0.6488 | **26.74** | 0.5807 |
| | PDw | | | | | | |
| | GTV | 35.44 | 0.8173 | 29.53 | 0.7239 | 25.86 | 0.6392 |
| | CZ | 35.21 | 0.8013 | 28.84 | 0.6632 | 24.67 | 0.5761 |
| | Yuan | 35.46 | 0.8174 | 29.29 | 0.7053 | 25.48 | 0.5915 |
| | SVHOVM | **36.82** | **0.8498** | **30.22** | **0.7246** | **26.15** | **0.6456** |
| | T2w | | | | | | |
| | GTV | 35.63 | 0.8338 | 28.51 | 0.7222 | 24.56 | 0.6082 |
| | CZ | 35.37 | 0.8167 | 27.89 | 0.6771 | 23.54 | 0.5923 |
| | Yuan | 35.64 | 0.834 | 28.25 | 0.7275 | 24.28 | 0.6201 |
| | SVHOVM | **36.65** | **0.8581** | **29.3** | **0.7415** | **24.74** | **0.655** |

the best PSNR measures. Following *Getreuer (2012)*, *Glowinski, Pan & Tai (2016)*, the penalty parameters $\theta_1$ and $\theta_2$ are set by 5 and 5, respectively. The number of iterations is fixed to 500. Beyond this, the performance of models is nearly unchanged.

Table 1 shows the PSNR and SSIM results of different models on the IXI and SB datasets. The figures highlighted in bold represent the best results for each volume and noise level. From Table 1, the following observations are made. First, the proposed model attains the best results for most of the cases. SVHOVM fails to achieve the best performance in SSIM for the T1w volume of the SB dataset with $\sigma = 15$ and 25. On average, SVHOVM outperforms GTV, CZ, and Yuan models by $0.76, 1.39, 0.95$ in PSNR and $0.0232, 0.0561, 0.00346$ in SSIM, respectively. Second, as the noise level increases, the improvement of SVHOVM over competitive models declines. The proposed model gives the average gains in PSNR of $1.23, 0.99,$ and $0.87$ for $\sigma = 5, 15,$ and 25, respectively; the corresponding figures in SSIM are $0.0349, 0.0233,$ and $0.0561$. It can be explained that large noises reduce the effectiveness of the weighting function. Third, SVHOVM yields the best performance on the T1w volume, followed by the PDw volume and then the T2w volume.

The advantages of SVHOVM are further confirmed by Figs. 6–8. Figure 6 shows denoising results of different models on an image of the IXI dataset with $\sigma = 5$. The parts of the denoised images are enlarged for visual comparison. Besides, the curves of 1D intensity values are shown. One can see from Fig. 6 that SVHOVM yields the best denoised image. For the results of SVHOVM, the flat regions are smoother and the object edges are

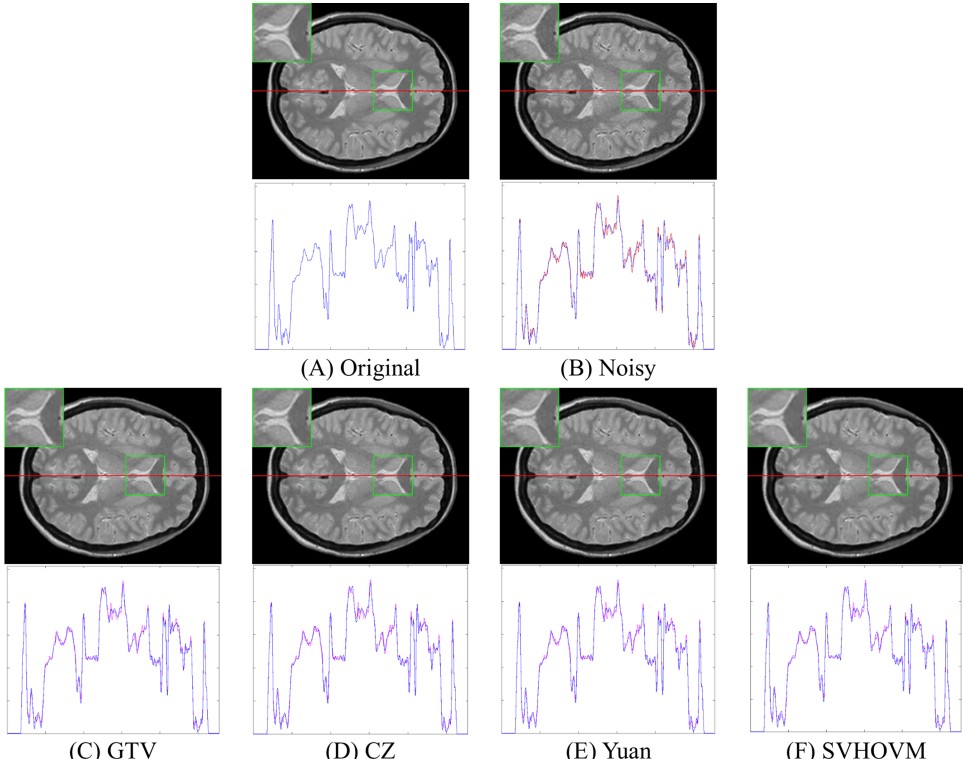

**Figure 6** **Denoising results of different models on an image of the IXI dataset with the noise level $\sigma =$ 5.** The parts of the denoised images, which are framed by green boxes, are enlarged for visual comparison. The 1D curves of intensity value are shown below the denoised images. Image source credit: IXI dataset, CC BY-SA 3.0 (https://brain-development.org/ixi-dataset/).

sharper compared with those of other models. The intensity curve of SVHOVM fits to the ground truth better than those of the competitive models.

Figures 7 and 8 demonstrate the restored images of different models on the images of the IXI dataset for the higher noise levels. The residual images, which are the image difference between the noisy images and the denoised images, are shown. It can be seen that the structural information exists in the residual images of all models. It is due to the fact that the Rician noise is signal-dependent. One can observe that fewer information is left in the residual images of SVHOVM over the cases of the competitive models. In summary, the quantitative and qualitative results show superior performance of SVHOVM compared with other existing variational models for Rician noise removal.

## CONCLUSION

In this article, the author presented a spatially variant high-order variational model for Rician noise removal. The SVTV regularizer was proposed in order to adjust the smoothing strength according to the characteristics of pixels. In addition, the proposed model employs the BH regularizer to reduce the staircase effect. The split Bregman algorithm was derived

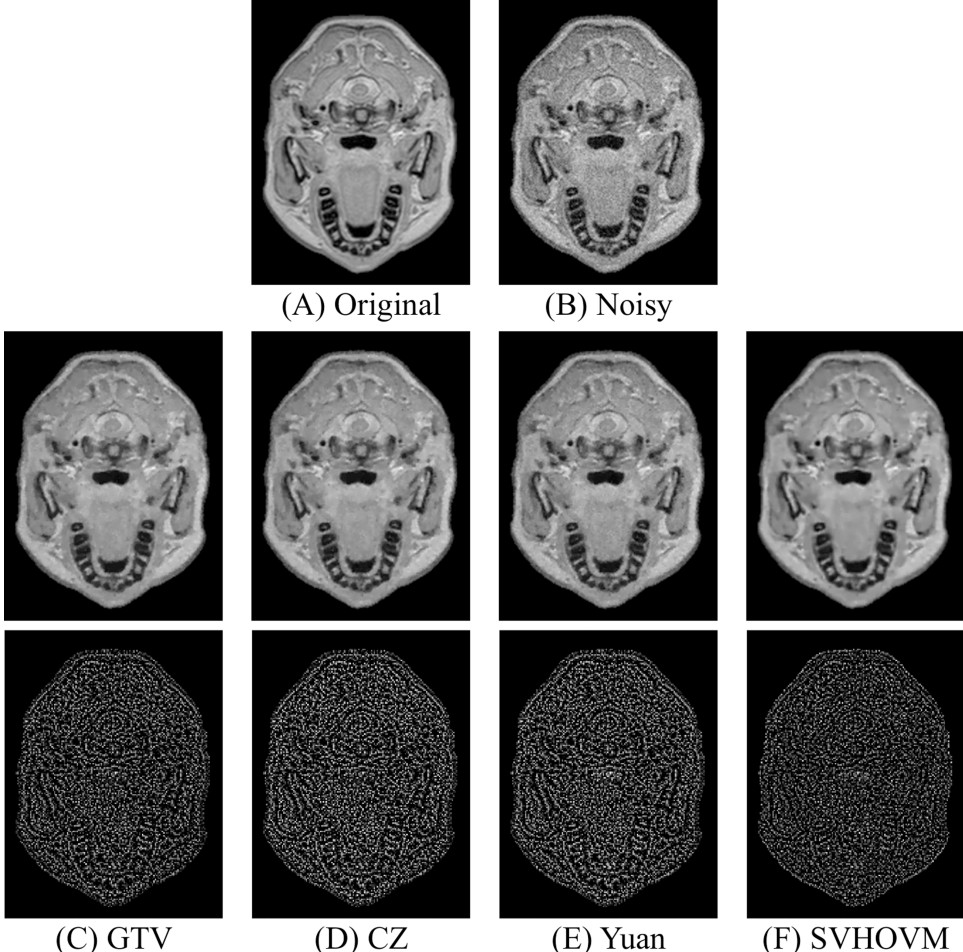

(A) Original    (B) Noisy

(C) GTV    (D) CZ    (E) Yuan    (F) SVHOVM

**Figure 7** Denoising results of different models on an image of the IXI dataset with the noise level $\sigma =$ 15 and the associated residual images Image source credit: IXI dataset, CC BY-SA 3.0 (https://brain-development.org/ixi-dataset/).

to solve the minimization problem efficiently. Extensive numerical experiments showed that the proposed model outperforms the existing variational models in terms of both quantitative and qualitative criteria.

The author hopes that the proposed method can serve as a tool for clinical analysis. One limitation of the proposed method is that it depends on parameters, especially the regularization parameters. On the one hand, the parameters allow the clinical experts to adjust the level of noise reduction to observe the image details. On the other hand, a parameter-dependent method requires the users to understand the effects of the parameters in order to obtain the optimal results. In the future, the author will investigate a method to automatically search for the optimal parameters of the proposed model.

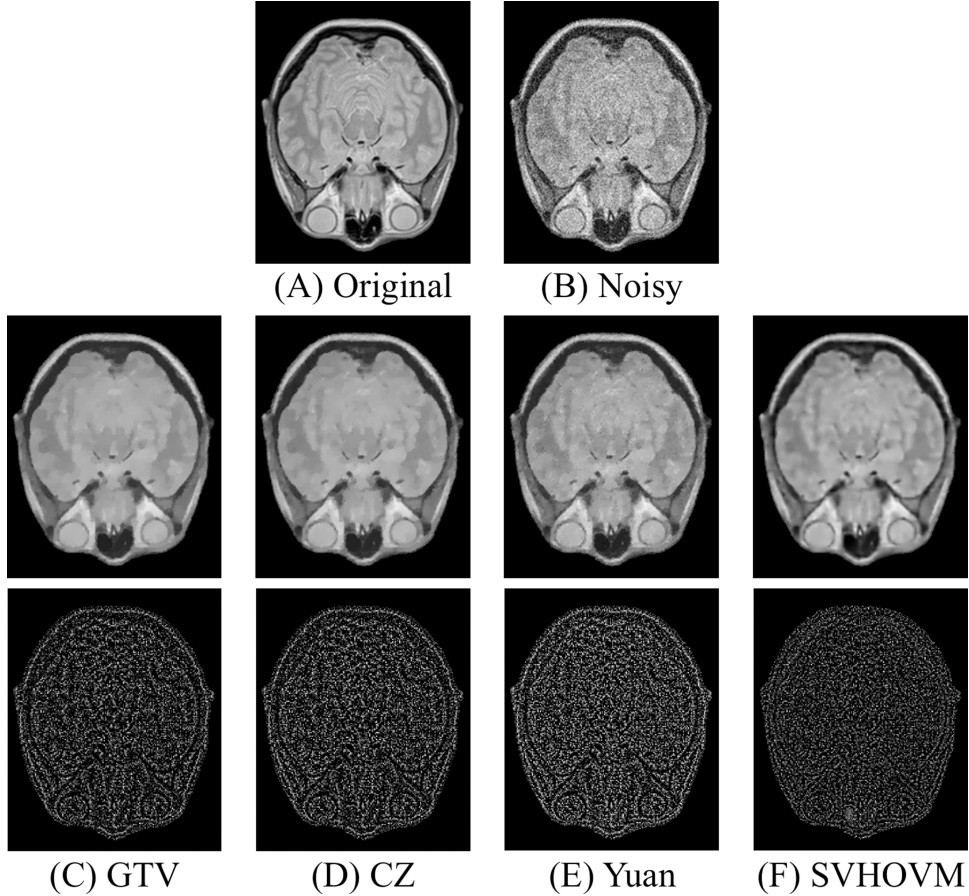

**Figure 8** Denoising results of different models on an image of the IXI dataset with the noise level $\sigma = 25$ and the associated residual images. Image source credit: IXI dataset, CC BY-SA 3.0 (https://brain-development.org/ixi-dataset/).

### Funding

This work was supported by the University of Danang, University of Science and Technology, code number of Project: T2022-02-36. The funders had no role in study design, data collection and analysis, decision to publish, or preparation of the manuscript.

### Grant Disclosures

The following grant information was disclosed by the author:
The University of Danang, University of Science and Technology: T2022-02-36.

### Competing Interests

The authors declare there are no competing interests.

## Author Contributions

- Tran Dang Khoa Phan conceived and designed the experiments, performed the experiments, analyzed the data, performed the computation work, prepared figures and/or tables, authored or reviewed drafts of the article, and approved the final draft.

## Data Availability

The data are available at BrainWeb: https://brainweb.bic.mni.mcgill.ca/brainweb/

The IXI dataset is available at Imperial College London: https://brain-development.org/ixi-dataset/

The code is available in the Supplemental Files.

## Supplemental Information

Supplemental information for this article can be found online at http://dx.doi.org/10.7717/peerj-cs.1579#supplemental-information.

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
