# Peer review of "A spatially variant high-order variational model for Rician noise removal"

_PeerJ Computer Science, doi:10.7717/peerj-cs.1579_

## Round 0.1 · original submission · Major Revisions

· Academic Editor

Major Revisions

The reviewers have substantial concerns about this manuscript. The authors should provide point-to-point responses to address all the concerns and provide a revised manuscript with the revised parts being marked in different color.

Reviewer 1 ·

Basic reporting

The proposed spatially variant high-order variational model (SVHOVM) introduces novelty into the field of Rician noise reduction. The use of a spatially variant TV regularizer that adjusts the smoothing strength per pixel based on its characteristics, coupled with the bounded Hessian (BH) regularizer to diminish the staircase effect, is an innovative approach to this problem. However, there are some parts to enhance whole paper quality.
1.It's crucial to maintain clarity and precision when presenting mathematical models in scientific literature. While reading through your equations, I noticed some notations that appear to be either missing or unclear. I recommend a thorough review of your equations to ensure all variables and constants are correctly defined and notated. For example, equation 13,

Experimental design

except figure 1 the flowchart. Here's a suggestion if you can create a graph that could demonstrate the cooperation between a spatially variant TV regularizer and bounded Hessian (BH) regularizer.

Validity of the findings

1.line 181. It's encouraging to see that SVHOVM outperformed other models on the SB dataset. However, to further strengthen the validity and generalizability of these findings, it would be beneficial to test SVHOVM on additional datasets. Specifically, the application of this method on Radiology dataset could provide valuable insight, considering the crucial role that MR imaging plays in cancer diagnosis and treatment.

2.For Figures 4 and 5, it would be informative to delve deeper into the impact of different x-axis parameter settings on y-axis outcomes. This can elucidate how varying these parameters influences the performance of the SVHOVM model. Such discussions could help in understanding the sensitivity of the model to parameter variations and the optimal settings for achieving the best performance.

3.In the conclusion section, further information about the potential usage of SVHOVM would be valuable. While it's clear that it has potential for MR imaging noise reduction, elaborating on specific scenarios, potential benefits in clinical or research settings, and its performance relative to other methods would enhance the paper's relevance and impact.

Additional comments

NA

Cite this review as

Reviewer 2 ·

Basic reporting

1. Figure 6, The difference between each column is not very clear, please zoom in and make the difference to be more readable.

Experimental design

1. Page 3, line 91, please provide the equation for calculating gamma function.

Validity of the findings

1. Figure 3, panel f-h. Images are provided with alpha_0 = 50 which is not shown in panel c-e. Please provide more details regarding why select alpha_0 = 50 or change it to images with alpha shown in label c-e.

Additional comments

1. Please also provide more discussions regarding the limitations of this study and future direction.

Cite this review as

---

## Round 0.2 · accepted · Accept

· Academic Editor

Accept

Reviewers are satisfied with the revisions, and I concur to accept this manuscript.

Reviewer 1 ·

Basic reporting

After reviewing the revised manuscript, I observed that the authors have diligently addressed all the comments and suggestions I provided earlier. Given the improvements and revisions made, I believe the paper is now in an acceptable state for publication.

Experimental design

N/A

Validity of the findings

N/A

Additional comments

N/A

Cite this review as

Reviewer 2 ·

Basic reporting

no comment

Experimental design

no comment

Validity of the findings

no comment

Additional comments

All the comments are answered properly, the manuscript can be published as is.

Cite this review as

---

## Author Rebuttal · Round 0.2

**University of Danang**

**University of Science and Technology**

[Figure]

*54 Nguyen Luong Bang, Danang, Vietnam*

*July 29th, 2023*

Dear Editor,

I appreciate the time and effort that the editor and the reviewers have dedicated to providing valuable feedback on my manuscript. The reviewers' remarks and suggestions helped me to significantly improve the manuscript. I have been able to incorporate changes to reflect most of the suggestions provided by the reviewers. To facilitate the work of the reviewers, I refer to the revised manuscript indicating the page and the line (P-L-).

Besides, I have also addressed all the technical changes in accordance with the requirements of the PeerJ's technical staffs about language, datasets, figure source credit, acknowledgements, figures.

I believe that the manuscript is now suitable for publication in PeerJ Computer Science.

Best regards,

Tran Dang Khoa Phan
* * *
## REVIEWER#1

### Basic reporting

The proposed spatially variant high-order variational model (SVHOVM) introduces novelty into the field of Rician noise reduction. The use of a spatially variant TV regularizer that adjusts the smoothing strength per pixel based on its characteristics, coupled with the bounded Hessian (BH) regularizer to diminish the staircase effect, is an innovative approach to this problem. However, there are some parts to enhance whole paper quality.

1. It's crucial to maintain clarity and precision when presenting mathematical models in scientific literature. While reading through your equations, I noticed some notations that appear to be either missing or unclear. I recommend a thorough review of your equations to ensure all variables and constants are correctly defined and notated. For example, equation 13.

**Author response:** The manuscript contains equations with many notations. We reviewed equations to ensure that all the notations are defined at the current or previous place. We added descriptions for some notations that are unclear (P3L91; P4L102,103). For Eq. 13, all the notations except $d$ and $z$ were defined in previous equations (Eqs. 1, 5, 11, 12).

### Experimental design

except figure 1 the flowchart. Here's a suggestion if you can create a graph that could demonstrate the cooperation between a spatially variant TV regularizer and bounded Hessian (BH) regularizer.

**Author response:** The SVTV regularizer has the ability of noise reduction but it generates the staircase effects. The BH regularizer is able to diminish the side effects produced by the SVTV term but it causes the image to be blurred. The denoising performance of our method depends on regularization parameters ($\alpha_0$ and $\beta$), which control the balance between the SVTV and the BH regularizers. The cooperation between them was demonstrated in Figs. 3 and 4A. Note that the order of figures was changed in this revised manuscript. We think that these figures are suitable to show the cooperation between these two regularizers.

### Validity of the findings

1.line 181. It's encouraging to see that SVHOVM outperformed other models on the SB dataset. However, to further strengthen the validity and generalizability of these findings, it would be beneficial to test SVHOVM on additional datasets. Specifically, the application of this method on Radiology dataset could provide valuable insight, considering the crucial role that MR imaging plays in cancer diagnosis and treatment.

**Author response:** We performed the additional evaluation on the IXI dataset, which is a public dataset of real MR images. The quantitative results and discussion were reported in Table 1, P8L179-P9L202. The visual results were shown in Figs. 6-8. Note that the order of figures was changed in this revised manuscript. The additional experiments strengthened the validity and generalizability of our method.

Besides, according to the requirements of the PeerJ's technical staffs, although the SB dataset generated by BrainWeb is public, it does not list any licensing or publication terms on the website. Meanwhile, the IXI dataset is public under a clear license (CC BY-SA 3.0). Thus, I had to remove all the images of the SB dataset and replace them with the images of the IXI dataset. Since MR images are quite similar, this adjustment does not affect the visual results. The quantitative results are kept unchanged. I was consulted by the PeerJ's technical staffs for all the changes related to the datasets.

2. For Figures 4 and 5, it would be informative to delve deeper into the impact of different x-axis parameter settings on y-axis outcomes. This can elucidate how varying these parameters influences the performance of the SVHOVM model. Such discussions could help in understanding the sensitivity of the model to parameter variations and the optimal settings for achieving the best performance.

**Author response:** We added more discussions for these figures (P8L151-156, P8L158-166).

3. In the conclusion section, further information about the potential usage of SVHOVM would be valuable. While it's clear that it has potential for MR imaging noise reduction, elaborating on specific scenarios, potential benefits in clinical or research settings, and its performance relative to other methods would enhance the paper's relevance and impact.

**Author response:** We added the discussion on the potential usage of SVHOVM as well as its limitations and the future plan (P9L204-P10L215).
* * *
**REVIEWER#2**

**Basic reporting**

1. Figure 6, The difference between each column is not very clear, please zoom in and make the difference to be more readable.

**Author response:** We edited Fig. 5 by zooming in to make the difference between columns more readable. Note that the order of figures was changed in this revised manuscript.

**Experimental design**

1. Page 3, line 91, please provide the equation for calculating gamma function.

**Author response:** We added the equation for the gamma function (P3L91).

**Validity of the findings**

1. Figure 3, panel f-h. Images are provided with alpha_0 = 50 which is not shown in panel c-e. Please provide more details regarding why select alpha_0 = 50 or change it to images with alpha shown in label c-e.

**Author response:** We fix $\alpha_0$ by a large value (50) in order to generate the staircase effect (see Fig. 3F). This side effect is not clear for small values of $\alpha_0$. Figs. 3F-H demonstrate that as $\beta$ gets larger, the BH regularizer diminishes the staircase effect more effectively, producing smooth transition between flat regions. We chose $\alpha_0 = 50$ instead of 100 because we did not

want to repeat Fig. 3E. So, we have 3 images (Figs. 3F-H), which are not repeated, to show the effect of $\beta$.

Note that according to the requirements of the PeerJ's technical staff, although the SB dataset generated by BrainWeb is public, it does not list any licensing or publication terms on the website. Meanwhile, the IXI dataset is public under a clear license (CC BY-SA 3.0). Thus, I had to remove all the images of the SB dataset and replace them with the images of the IXI dataset. Since MR images are quite similar, this adjustment does not affect the visual results. Although the values of parameters $\alpha_0$ and $\beta$ in Fig. 3 were changed, the goal of this figure is the same as described above. I was consulted by the PeerJ's technical staffs for all the changes related to the datasets.

## Additional comments

1. Please also provide more discussions regarding the limitations of this study and future direction.

**Author response:** We added the discussion on the potential usage of SVHOVM as well as its limitations and the future plan (P10L212-217).